# High efficiency and long-term intracellular activity of an enzymatic nanofactory based on metal-organic frameworks

Xizhen Lian[1], Alfredo Erazo-Oliveras[2], Jean-Philippe Pellois[1,2] & Hong-Cai Zhou [1]

Enhancing or restoring enzymatic function in cells is highly desirable in applications ranging from ex vivo cellular manipulations to enzyme replacement therapies in humans. However, because enzymes degrade in biological milieus, achieving long-term enzymatic activities can be challenging. Herein we report on the in cellulo properties of nanofactories that consist of antioxidative enzymes encapsulated in metal–organic frameworks (MOFs). We demonstrate that, while free enzymes display weak activities for only a short duration, these efficient nanofactories protect human cells from toxic reactive oxygen species for up to a week. Remarkably, these results are obtained in spite of the nanofactories being localized in lysosomes, acidic organelles that contain a variety of proteases. The long-term persistence of the nanofactories is attributed to the chemical stability of MOF in low pH environment and to the protease resistance provided by the protective cage formed by the MOF around the encapsulated enzymes.

---

[1] Department of Chemistry, Texas A&M University, College Station, TX 77843-3255, USA. [2] Department of Biochemistry and Biophysics, Texas A&M University, College Station, TX 77843-2128, USA. Correspondence and requests for materials should be addressed to J.-P.P. (email: pellois@tamu.edu) or to H.-C.Z. (email: zhou@chem.tamu.edu)

The modulation of protein function in live cells is valuable in biotechnology and medicine[1]. Rescuing enzymatic activity can for instance provide therapeutic benefits for the many diseases associated with defective enzymes[2]. A general approach used to restore protein function involves genetic manipulation, whereby introduction of a gene into cells replaces its defective counterpart and insures the production of its protein product otherwise lacking. However, genetic manipulations are often problematic as they may inadvertently alter the genome of cells and lead to new diseases, including cancer[3–7]. To circumvent this issue, directly using protein supplementation may be preferable. However, proteins, in part due to their relative large size and hydrophilicity, do not readily penetrate cells. Proteins also have relatively short extra or intracellular half-lives. If not produced directly inside a cell, proteins may therefore not reach the location where their activity is required, and, additionally, not produce a prolonged effect because of their rapid degradation.

Addressing the problems associated with protein replacement has been the focus of intense research. Several techniques have been developed to enhance the cellular delivery of these macromolecules, for instance, protein PEGylation or encapsulation in liposomes, micelles or polymersomes, to improve transport properties and increase protein half-lives[8–18]. Another key idea is to increase protein stability by protecting these macromolecules from proteolytic degradation using encapsulation agents. In addition, encapsulation can help prevent immunological responses that may occur from exogenously introduced proteins[9,10,19]. These strategies have been successful in enhancing protein retention time in the circulatory system and in reducing undesired accumulation in the liver[19–21]. However, other challenges remain. For instance, encapsulated enzymes are often quiescent until they are released from their carrier and, because release is often inefficient, only a small fraction of the total available enzymatic activity is displayed at one time[10]. Surface modifications with chemical moieties such as PEG can also

significantly alter protein structure and reduce activity[22,23]. Overall, developing technologies that protect proteins from degradation while maintaining optimal protein function is presently highly desirable.

A possible solution to the problems currently associated with protein formulation may lie in the recently developed structures known as metal–organic frameworks (MOFs). MOFs are an emerging type of porous materials constructed from metal containing clusters and organic linkers. Due to the high porosity as well as structural and functional tunability, MOFs hold promises in a variety of applications, including gas storage/separation, catalysis, and sensing[24–30]. Recently, enzymes have been loaded into the cavities of MOFs and immobilized enzymes tested thus far (e.g., horseradish peroxidase (HRP), cytochrome c (Cyt c), etc.) have displayed robust in vitro activities[31–36]. This indicates that proteins can fold properly in the cavities of MOFs and remain functionally active. MOF-immobilized enzymes have also shown extraordinary stabilities under denaturing conditions such as extreme heat, high or low pH, and in the presence of organic solvents[37–39]. Moreover, the cage formed by MOFs acts as a barrier against proteases, such as trypsin, and protects encapsulated proteins from proteolytic degradation[40,41].

The properties displayed in vitro by MOF-enzyme nanofactories are highly attractive. Although the biocompatibility of some MOF materials have been investigated in a number of reports, whether MOF-enzyme composites may serve as efficient nanofactories in living cells remains untested[42–52]. Herein, we aimed to test the hypothesis that intracellular MOF nanofactories are capable of supporting an enzymatic activity beneficial to living cells for an extended period of time. To test this hypothesis, we chose PCN-333(Al) as a MOF platform due to its ultrahigh enzyme encapsulation capacity, facile fluorescence modification, and excellent chemical robustness in aqueous solutions[53]. As a proof-of-concept study, we established that PCN-333 based nanofactories containing the antioxidant enzymes, superoxide

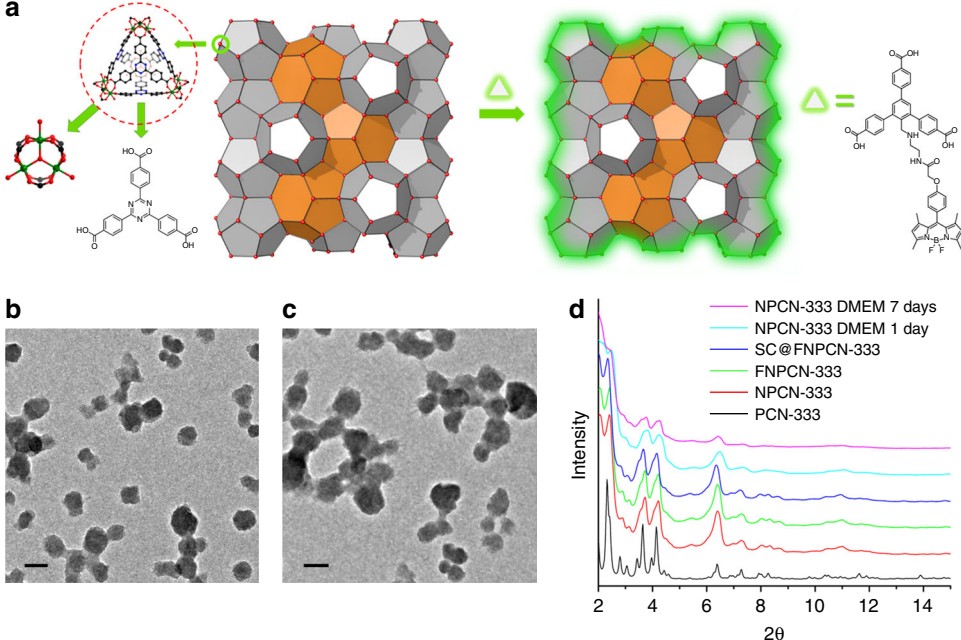

**Fig. 1** Structure and characterization of PCN-333 nanoparticles. **a** PCN-333 is composed of trimeric clusters and TATB ligands, which self-assemble into supertetrahedra (highlighted in red dashed circle). The supertetrahedra are connected with each other in a vertex-sharing manner. A green fluorophore is anchored on NPCN-333 via ligand metathesis. **b** TEM image of NPCN-333. Scale bar: 100 nm. **c** TEM image of FNPCN-333. Scale bar: 100 nm. **d** Powder X-ray diffraction patterns (2θ from 2 to 15 degree) of microscale PCN-333 (black); NPCN-333 (red); FNPCN-333 (green); SC@FNPCN-333 (blue); NPCN-333 soaked in DMEM for 1 day (cyan); NPCN-333 soaked in DMEM for 7 days (magenta)

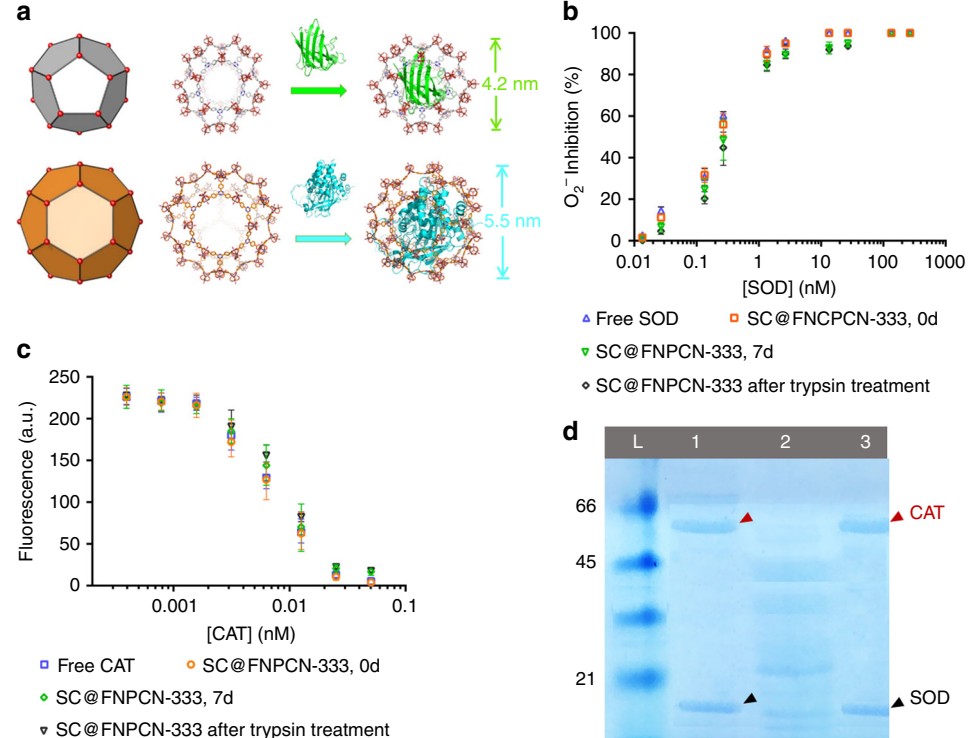

**Fig. 2** Enzyme encapsulation and in vitro determination of enzymatic activities. **a** Schematic representation illustrating the relative size of the cavities present in FNPCN-333 and of the enzymes SOD and CAT (from PDB 1CBJ and 5GKN). The medium (gray) and large (orange) cavities of FNPCN-333 accommodate SOD (green) and CAT (cyan), respectively. **b** Relative enzymatic activities of several SOD formulations, as determined by the superoxide inhibition assay: free SOD (blue), freshly prepared SC@FNPCN-333 (orange), SC@FNPCN-333 stored for 7 days (green), SC@FNPCN-333 treated with trypsin for 2 h (black). $n = 3$, mean ± s.d. **c** Relative enzymatic activities of several CAT formulations, as measured by the inhibition of $H_2O_2$-mediated production of the fluorophore resorufin: free CAT (blue), freshly prepared SC@FNPCN-333 (orange), SC@FNPCN-333 stored for 7 days (green), SC@FNPCN-333 treated with trypsin for 2 h (black). $n = 3$, mean ± s.d. **d** SDS-PAGE analysis of SC@FNPCN-33 after exposure to the protease trypsin. L: protein ladder. Lane 1: free SOD (highlighted with black arrow) and CAT (highlighted with red arrow). Lane 2: Free SOD and CAT after treatment with trypsin for 1 h. Lane 3: SC@FNPCN-333 treated with trypsin for 1 h (analyzed after dissolution of MOF in HCl)

dismutase (SOD) and catalase (CAT), protect cells from severe oxidative stress, a process involved in a large number of pathological states[54–57]. Remarkably, this protective effect was maintained for a minimum of a week despite the localization of the MOFs inside lysosomes, one of the most degradative environments in living cells[58].

## Results

**Preparation and characterization of PCN-333 nanoparticles.** The basic secondary building unit of PCN-333 is a supertetrahedron, which consists of an aluminum trimeric cluster at the four vertices and TATB ligands on the faces, in a vertex-sharing manner (Fig. 1a). The supertetrahedron is the secondary building block for two mesoporous cavities: a smaller dodecahedral cage composed of 20 supertetrahedra connected by vertex sharing with exclusive pentagonal windows and a larger hexacaidecahedral (hexagonal-truncated trapezohedral) cage surrounded by 28 supertetrahedra with both pentagonal and hexagonal windows (Fig. 2a). The cavity size of supertetrahedral, dodecahedral and hexacaidecahedral cage is 1.1, 4.2 and 5.5 nm. Nanoscale PCN-333 (NPCN-333) was synthesized in the condition similar to that of microscale PCN-333, with differences lying in the concentration of starting material and the amount of modulating reagents. Transmission electron microscopy (TEM) images indicated that NPCN-333 particles possessed spherical shape with an average diameter of 100 nm (Fig. 1b). This diameter was coincident with that measured by dynamic light scattering (DLS) (Supplementary Fig. 1). The as-synthesized particles demonstrated high

crystallinity as determined by powder X-ray diffraction (PXRD) (Fig. 1d). The pattern was isostructural to PCN-333, although the peaks were broadened due to the reduced particle size. $N_2$ isotherm at 77 K revealed that the NPCN-333 was highly porous with hierarchical porosity (Supplementary Fig. 5). The surface area was 2793 m$^2$ g$^{-1}$ and the void volume was 2.94 cm$^3$ g$^{-1}$. FNPCN-333, a fluorescent version of NPCN-333 prepared for live cell fluorescence microscopy experiments, was prepared via ligand metathesis of NPCN-333 with a BTB ligand functionalized with the fluorophore BODIPY (Fig. 1a). FNPCN-333 displays particle size, distribution, and porosity properties similar to those of NPCN-333 (surface area 2428 m$^2$ g$^{-1}$, void volume 2.30 cm$^3$ g$^{-1}$) (Fig. 1c, Supplementary Fig. 7). The presence of BTB-Green on the framework backbone is also confirmed by the relatively larger distribution of microporosity in FNPCN-333 than that of NPCN-333 (Supplementary Fig. 9). However, unlike NPCN-333, FNPCN-333 is fluorescent with a maximal emission at 509 nm in water (Supplementary Fig. 10). As previously reported, PCN-333 is stable in aqueous solutions over a pH range of 3 to 9. When particle size decreased, the chemical stability was not compromised. The well maintained crystallinity of FNPCN-333 was confirmed by the unchanged PXRD pattern (Fig. 1d) as well as the fringes on the crystals (Supplementary Fig. 11) after soaking in the cell culture media for up to 7 days.

**Enzyme encapsulation and protection by FNPCN-333.** An early examination of the molecular dimensions of SOD ($2.8 \times 3.5 \times 4.2$ nm$^3$, 16.3 kDa)[59] and CAT ($4.9 \times 4.4 \times 5.6$ nm$^3$,

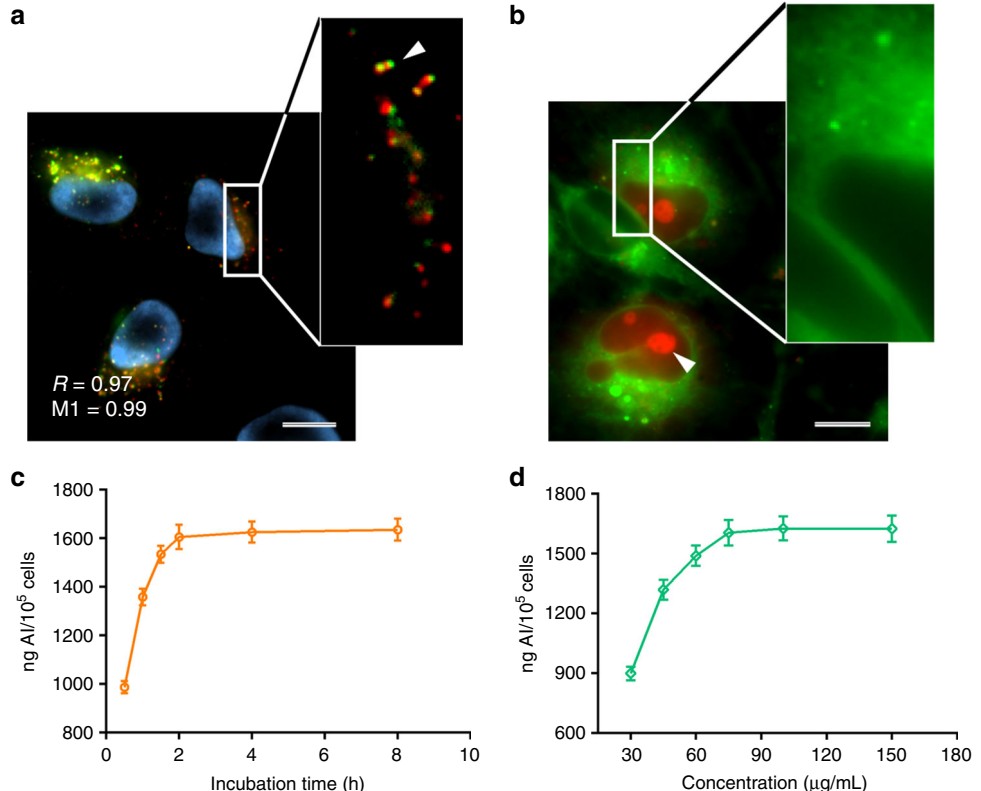

**Fig. 3** Cellular internalization of SC@FNPCN-333 by HeLa cells. **a** 100× microscopy images of SC@FNPCN-333 (75 μg mL$^{-1}$) incubated with HeLa cells for 2 h, then stained with the nuclear marker Hoechst 33,342 and LysoTracker red. Cell nuclei, SC@FNPCN-333 and LysoTracker red labeled acidic organelles (i.e. late endosomes and lysosomes) are pseudocolored in cyan, green and red, respectively. The arrow in the enlarged green/red overlay image illustrates the co-localization of puncta containing both SC@FNPCN-333 and LysoTracker red. Pearson's correlation coefficient R and Mander's correlation coefficient M1 is indicated[79,80]. Scale bar: 10 μm. **b** 100× microscopy images of SC@FNPCN-333 co-incubated with the cytosolic delivery agent dfTAT for 2 h. The image is an overlay of SC@FNPCN-333 (pseudocolored green) and dfTAT (pseudocolored red). The arrow points to a dfTAT-stained nucleolus, indicative of effective cytosolic penetration of the delivery reagent. The enlarged image illustrates the green fluorescence of SC@FNPCN-333 in the cytosolic space and, by contrast, a dark area suggestive of the exclusion of SC@FNPCN-333 from the cell nucleus. Scale bar: 10 μm. **c** ICP-MS analysis of Al content in HeLa cells incubated with SC@FNPCN-333 (75 μg mL$^{-1}$) for varying time points. $n = 3$, mean ± s.d. **d** ICP-MS analysis of Al content in HeLa cells incubated with varying concentrations of SC@FNPCN-333 for 2 h. $n = 3$, mean ± s.d

60 kDa)[60] indicated that these proteins could potentially fit in the 4.0 and 5.5 nm cavities of FNPCN-333, respectively. To test this idea, the two enzymes were incorporated into FNPCN-333 in a stepwise manner[41]. First, FNPCN-333 was incubated with CAT so as to occupy the larger MOF cavities. After addition of CAT, FNPCN-333 was incubated with SOD to load the smaller cavities that would not accommodate CAT. Based on BCA analysis, the encapsulation capacity of FNPCN-333 for SOD and CAT was 0.80 and 1.26 g g$^{-1}$, respectively. This is comparable to the calculated values of maximal encapsulation capacity (0.92 and 1.74 g g$^{-1}$) and is indicative of a high loading efficiency. Consistent with this notion, N$_2$ isotherm analysis shows that enzyme encapsulation leads to complete disappearance of the two mesoporous cavities (Supplementary Fig. 15). In contrast, the microporous cavities that are too small in volume to accommodate the protein molecules remain detectable. The crystallinity of the resulting bi-enzymatic nanofactory, named SC@FNPCN-333, was well maintained, as indicated by PXRD patterns. Thermal gravimetric analysis (TGA) further confirms high enzyme loading of FNPCN-333 (Supplementary Fig. 17).

In order to determine whether encapsulated SOD and CAT are biologically functional, the water soluble tetrazolium (WST) and horseradish peroxidase (HRP)-Amplex Red assays were performed. In the WST assay, the activity of SOD is detected by spectroscopically measuring the superoxide-mediated reduction

of WST (pale yellow) to the formazan dye (dark yellow). In the HRP-Amplex Red assay, the fluorescent dye resorufin is produced upon oxidation of Amplex Red with hydrogen peroxide, a reaction catalyzed by HRP. By catalyzing the decomposition of hydrogen hydroxide, CAT inhibits the generation of resorufin. As shown in Fig. 2b, the superoxide detoxifying activity of SC@FNPCN-333 was comparable to that of free SOD. Similarly, the peroxide scavenging activity of SC@FNPCN-333 was comparable to that of free CAT (Fig. 2c). Notably, SC@FNPCN-333 displayed persistent CAT and SOD enzymatic activities after soaking in the cell culture media (DMEM) for 7 days.

Resistance of the enzymatic nanofactory towards cellular degradation, from factors such as proteases and acidic pH, is crucial for its long-term performance in living cells. To test whether NPCN-333 could provide a protective environment for SOD and CAT, SC@FNPCN-333 was first exposed in vitro for 2 h to the protease trypsin. SC@FNPCN-333 was subsequently treated with HCl so as to dissolve the MOF and release encapsulated proteins, including potential trypsin-digested enzyme fragments. The resulting supernatants were analyzed by SDS-PAGE. Trypsin readily digests SOD and CAT in their free form, as illustrated by the presence of small molecular weight bands (Fig. 2d). In contrast, bands corresponding to intact SOD and CAT are predominantly present for SC@FNPCN-333. In

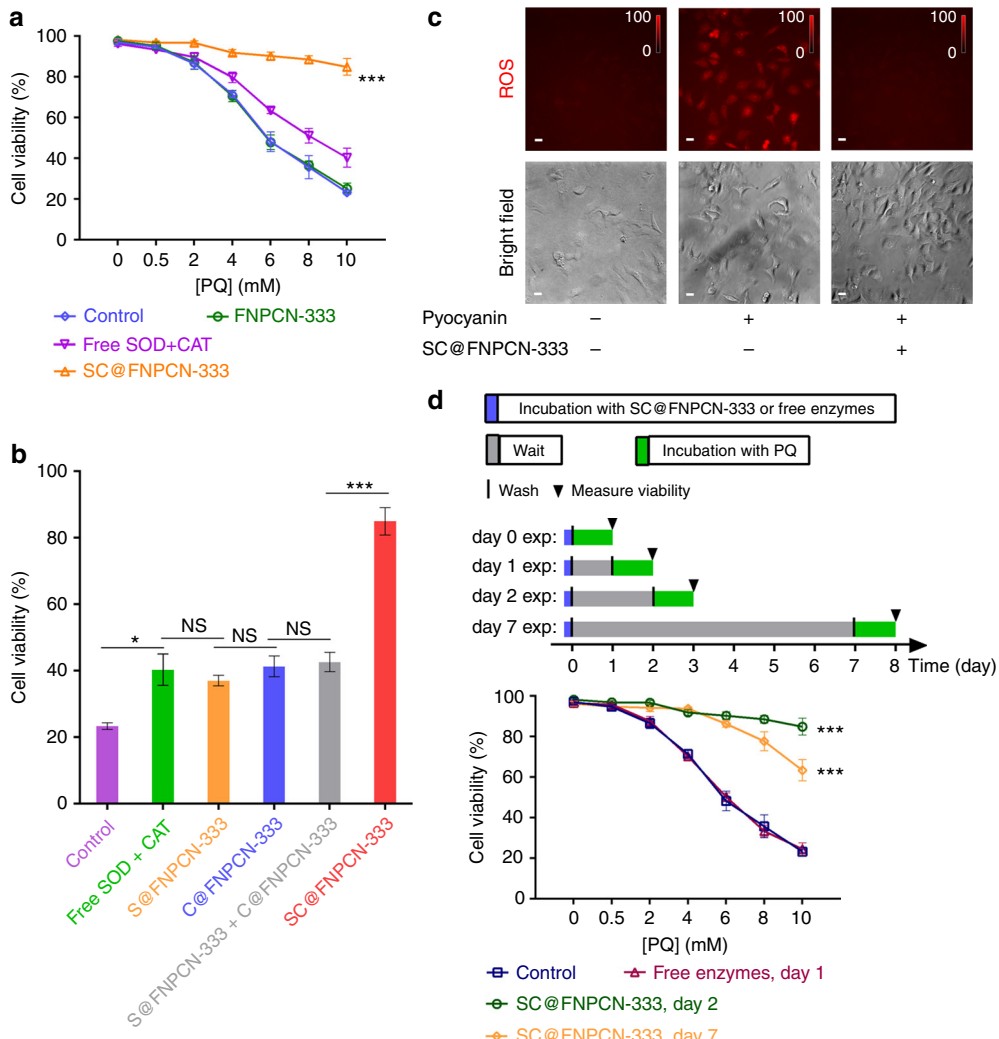

**Fig. 4** Protection of HeLa cells from oxidative stress by SC@FNPCN-333. **a** Viability of cells treated with the oxidant paraquat (PQ) for 24 h. Blue: control. Green: cells pre-treated with FNPCN-333. Orange: cells pre-treated with SC@FNPCN-333 (75 μg mL$^{-1}$) for 2 h. Magenta: cells pre-treated with SOD (60 μg mL$^{-1}$) and CAT (94.5 μg mL$^{-1}$) for 2 h. $n = 5$, mean ± s.d. *** represents $P \leq 0.001$. **b** Viability of cells treated with PQ (10 mM) for 24 h following pre-treatments with various enzyme formulations. Purple: control of cells with no pretreatment. Green: cells pre-treated with SOD (60 μg mL$^{-1}$) and CAT (94.5 μg mL$^{-1}$) for 2 h. Orange: cells pre-treated with S@FNPCN-333 (75 μg mL$^{-1}$) for 2 h. Blue: cells pre-treated with C@FNPCN-333 (75 μg mL$^{-1}$) for 2 h. Light gray: cells pre-treated with S@FNPCN-333 (75 μg mL$^{-1}$) and C@FNPCN-333 (75 μg mL$^{-1}$) for 2 h. Red: cells pre-treated with SC@FNPCN-333 (75 μg mL$^{-1}$) for 2 h. $n = 5$, mean ± s.d. *$P < 0.05$ ***$P \leq 0.001$, respectively. **c** In cellulo ROS detoxification by SC@FNPCN-333. Presence of superoxide, which generates an oxidized and red fluorescent form of SDR, was detected by fluorescence microscopy. Fluorescence microscopy images (20× magnification) of SDR are pseudocolored red. Images were acquired and processed using identical conditions. Intensity bars show the relative fluorescence intensities of the 3 images (normalized to a highest signal of 100). The experiment was replicated for three times and representative images are displayed. Scale bar: 10 μm. **d** Long-term persistence of the protective effect of SC@FNPCN-333. Cells were pretreated with SC@FNPCN-333 (75 μg mL$^{-1}$, 2 h), washed and cultured for several days (gray time period in scheme). Cells were then treated with PQ (24 h incubation) and cell viability was quantified. Dark blue: control of cells not pretreated with SC@FNPCN-333. Dark magenta: cells pre-treated with SOD (60 μg mL$^{-1}$) and CAT (94.5 μg mL$^{-1}$), PQ-protection assessed 24 h post-treatment. Green: cells pre-treated with SC@FNPCN-333 (75 μg mL$^{-1}$), PQ-protection assessed 48 h post-treatment. Orange: cells pre-treated with SC@FNPCN-333 (75 μg mL$^{-1}$), PQ-protection assessed 168 h post treatment. $n = 5$, mean ± s.d. ***$P \leq 0.001$

addition, SC@FNPCN-333 maintained its enzymatic activities after trypsin treatment, as indicated by the WST and HRP-Amplex Red assays (Fig. 2b and c). In order to test whether the nanofactories resist exposure to an acidic milieu, SOD, CAT and SC@FNPCN-333 were incubated at pH 5 (this pH mimics the luminal pH of late endosomes and lysosomes, organelles inside which MOF can accumulate, as shown below). Enzymatic activities were measured at 0.5 and 24 h, and compared to those obtained at pH 7.4. After 0.5 h incubation, the acidic pH led to a moderate loss in activity for the free enzymes and for SC@FNPCN-333 (approximately 15 to 25% for SC@FNPCN-333

and CAT, 35% for SOD) (Supplementary Figs. 19 and 20). After 24 h incubation, the activities of SOD and CAT at pH 5 were reduced to only 10 or 20% of those obtained at pH 7.4, presumably because of slow unfolding of the proteins. In contrast, the activities of SC@FNPCN-333 at pH 5 were unchanged during the course of the experiment (>80% relative activity at 1 and 24 h) (Supplementary Figs. 21 and 22). Altogether, these experiments indicate that FNPCN-333 provides a platform for high enzymatic activity, protection from proteases, and resistance under a broad pH range. These encouraging data also set the stage for in cellulo testing.

**Cellular uptake of SC@FNPCN-333**. In order to exert optimal enzymatic activities, exogenously administered enzyme nanofactories should ideally reach the intracellular locations occupied by their endogenous counterparts. SOD is localized in the cytosolic space and mitochondria of human cells while CAT is typically localized in peroxisomes[61,62]. On one hand, given the size and biophysical properties of MOF, we did not anticipate that SC@FNPCN-333 would spontaneously penetrate into the cytosolic space of cells and reach these intracellular destinations. On the other hand, 100 nm particles are small enough to allow for cellular uptake by endocytosis. We therefore envisioned that endosomal entrapment of SC@FNPCN-333 should be possible. Furthermore, we have recently developed dfTAT, a delivery reagent that causes leakage of late endosomes and permits the release of endocytosed cargos into the cytosolic space of cells[63–65]. To test whether SC@FNPCN-333 endosomal or cytosolic localization could be achieved, SC@FNPCN-333 was incubated with HeLa cells for 2 h, with or without dfTAT. When administered alone to live cells, SC@FNPCN-333 displayed an intracellular punctate fluorescence distribution, as observed by microscopy. Co-localization of SC@FNPCN-333 with LysoTracker red, a late endosome and lysosome marker[66], confirmed entrapment of the enzyme nanofactories within these endocytic organelles (L.E./LYS, Fig. 3a). Consistent with these results, incubation of SC@FNPCN-333 with HeLa cells at 4 °C, a condition that inhibits energy-dependent endocytic uptake, or in the presence of amiloride, an inhibitor of macropinocytosis, led to a dramatic reduction in the number of intracellular SC@FNPCN-333-positive puncta and to a reduction of co-localization with Lyso-Tracker red (Supplementary Fig. 23). When administered in the presence of dfTAT, SC@FNPCN-333 presented a homogeneously diffuse fluorescence distribution, indicative of localization in the cytosolic area, albeit was accompanied by a residual punctate distribution indicative of incomplete endosomal escape. Moreover, consistent with the notion that the size of SC@FNPCN-333 exceeds the nuclear pore complex size threshold for passive diffusion, the cytosolic green fluorescence distribution of SC@FNPCN-333 was excluded from nuclei (Fig. 3b). By extension, this staining supports the notion that the diffuse fluorescence observed from SC@FNPCN-333 is indeed cytosolic (in other words, artifactual out-of-focus extracellular signals would not be selectively excluded from nuclei). Overall, these results indicate that the intracellular accumulation of enzymes nanofactories can be achieved and that their intracellular distribution can be modulated. Notably, the cytotoxicity of SC@FNPCN-333, evaluated using a SYTOX Blue exclusion assay, was negligible, in the presence and absence of dfTAT. In particular, cell viability was more than 97% at 0 or 24 h post-incubation, a result identical to that obtained with untreated cells.

In order to optimize MOF cellular uptake, and given that the two cellular delivery approaches used above are dependent on endocytosis, we next tested the dependence of MOF endocytic uptake on time and concentration. To quantify the endocytic uptake of SC@FNPCN-333, the aluminum (Al) content of cells incubated with SC@FNPCN-333 was measured by ICP-MS. For instance, cells were incubated with 75 μg mL$^{-1}$ SC@FNPCN-333, harvested at different time points, and lysates were prepared for analysis. As shown in Fig. 3c, Al levels dramatically increased after 30 min incubation and, after 2 h incubation, reached a plateau corresponding to 89.2 ± 3.7 nmol nanoparticle per $10^5$ cells (Fig. 3c). Alternatively, cells were incubated with varying amounts of SC@FNPCN-333 and analyzed after 2 h incubation. In this case, internalization initially increased with SC@FNPCN-333 concentration but reached a plateau at 75 μg mL$^{-1}$ (Fig. 3d). Altogether, these results indicate that incubation of cells with 75 μg mL$^{-1}$ SC@FNPCN-333 for 2 h leads to maximal cellular

uptake. We therefore chose these conditions for subsequent studies.

**In cellulo activity of SC@FNPCN-333**. To address whether SC@FNPCN-333 can protect cells from ROS-induced toxicity, we established an assay in which cells are exposed to oxidative stress. In particular, we treated cells with paraquat (PQ), a redox-active herbicide previously linked to various human diseases, including organ failure and Parkinson's disease[67]. As expected, cells treated with varying concentrations of PQ (0.5–10 mM) for 24 h showed reduced cell viability when compared to untreated cells (Fig. 4a). Pretreating cells with FNPCN-333 (75 μg mL$^{-1}$, 2 h) did neither increase nor decrease cell viability, indicating that MOF alone do not interfere with the toxicity of PQ. In contrast, when cells were pre-treated with SC@FNPCN-333 (75 μg mL$^{-1}$, 2 h; this condition leads to L.E./LYS accumulation), cell viability was restored to a high degree, with more than 85% of cells surviving the most cytotoxic PQ treatment. In comparison, pretreating cells with either free enzymes, or with MOF loaded with only SOD or CAT (S@FNPCN-333 or C@FNPCN-333) led to only a minor improvement in cell viability. Moreover, a mixture of S@FNPCN-333 and C@FNPCN-333 did not perform as well as SC@FNPCN-333, pointing to a beneficial effect of enzyme proximity obtained from co-encapsulation (Fig. 4b). In order to validate that SC@FNPCN-333 exerts a protective effect by mediating ROS detoxification, the impact of the enzymes nanofactory on ROS levels was assessed by microscopy. Cells were incubated with pyocyanin, a bacterial toxin that produces superoxide, and a fluorescent Superoxide Detection Reagent (SDR). Microscopy imaging shows that, the fluorescence of SDR, which is proportional to the superoxide concentration in cells, increased in the presence of pyocyanin. However, upon pretreatment with SC@FNPCN-333, the SDR signal was restored to basal level (Fig. 4c). When the cellular uptake of SC@FNPCN-333 was inhibited by either low temperature incubation (4 °C) or amiloride, viability against PQ was barely improved compared to cells not pre-treated with SC@FNPCN-333. Together, these results confirm that SC@FNPCN-333 protects cells from ROS-induced toxicity by eliminating ROS species. Finally, similar PQ-protection results were obtained in the presence of dfTAT, indicating that differences in intracellular localization (cytosolic vs. endocytic) does not significantly impact the ROS detoxifying capacity of the enzyme nanofactories (Supplementary Fig. 29). Given that L.E./LYS are extremely degradative and potentially more damaging than the cytosolic space, we next tested the long-term performance of the enzymes nanofactories in this environment for maximal stringency.

To maximize their application and usefulness, enzyme nanofactories should have a persistent effect inside cells. To test whether SC@FNPCN-333 could sustain its protecting effect beyond 24 h, cells pre-incubated with SC@FNPCN-333 were cultured with fresh DMEM buffer for up to a week. PQ-induced toxicity was evaluated at day 2 and day 7. As shown in Fig. 4d, treating cells with free SOD and CAT, which provided only a weak protective effect at day 0, failed to rescue cells from PQ-induced cytoxicity at day 1. In contrast, the rescuing effect of SC@NPCN-333 at day 2 was similar to that obtained at day 1. Moreover, this effect, although partially diminished, remained significant at day 7. This long-term persistence is consistent with the stability of the MOF material detected in vitro, where little protein leaching out of the MOF is observed over 7 days in an acidic milieu (Supplementary Figs. 40 and 41). Notably, ICP-MS analysis shows that the intracellular aluminum content present at day 7 was approximately 6% of that of cells at day 0 (1600 ng Al per $10^5$ cells are detected after incubation of 75 μg mL$^{-1}$ of

SC@FNPCN-333 at day 0, 104 ng Al per $10^5$ cells at day 7; Supplementary Fig. 43). This is therefore indicative of a decline in overall MOF concentration of present in cells overtime, as would be expected from the dilution that take place during each cell division during 1 week of cell culture. However, titration experiments confirmed that lower doses of SC@FNPCN-333 could maintain high protection activities in cells. In particular, while incubation of cells at 1 µg mL$^{-1}$ SC@FNPCN-333 provides no protective effect, incubation with 2.5 µg mL$^{-1}$ SC@FNPCN-333 leads to approximately 100 ng Al per $10^5$ cells and 70% cell viability after PQ treatment (a condition similar to that obtained at day 7 after incubation of 75 µg mL$^{-1}$ SC@FNPCN-333, Supplementary Figs. 44 and 45). Overall, these results indicate that the prolonged effect of SC@FNPCN-333 is imparted by the chemical stability of the material as well as its sustained enzymatic activity even at low intracellular concentrations.

## Discussion

This report establishes that enzymatic nanofactories based on MOF can sustain intracellular enzymatic activities for an extended period of time. The proteins encapsulated within the MOF structure are enzymatically active, indicating that proper enzyme folding is achieved and maintained within the MOF and that soluble substrates and products can diffuse in and out of the MOF-enzyme nanofactories. The MOF act as a nanocage that protects encapsulated enzymes from proteases and the acidic environment inside L.E./LYS. The mechanism of protease resistance presumably involves restricting the access of a protease to its protein substrate, in this case CAT and SOD. In particular, while trypsin may be small enough to enter an empty MOF cavity, it is unlikely that it would be able to do so if the cavity is already loaded with SOD or CAT. Alternatively, the MOF environment may stabilize the folding/structure of encapsulated enzymes. This would also contribute to improving protease resistance (unstructured proteins are more readily proteolized than folded proteins) and could explain how the MOF prevents pH-induced unfolding and loss of activity. Overall, these features provide a material that persists in the lumen of L.E./LYS, a highly degradative cellular milieu that combines a high density of proteases and acidic pH. Based on our in vitro and in cellulo data, it is likely that free enzymes are rapidly degraded in these organelles. It is also likely that it is the slow leaching of enzymes from the MOF structure that contribute to the gradual loss of activity detected in live cells.

It is notable that the proximity of SOD and CAT in the enzymatic nanofactory enhances the protective effect against PQ-induced oxidative stress. SOD catalyzes the disproportionation of superoxide and generates $H_2O_2$ and oxygen, while $H_2O_2$ is consumed by CAT, yielding water and oxygen. Since the two reactions catalyzed by SOD and CAT may occur in a cascade manner, it is likely that SOD-generated $H_2O_2$ is degraded by CAT before this ROS diffuses away from the nanofactory. Conversely, in the case of MOF nanoparticles carrying only one enzyme, the $H_2O_2$ generated by S@FNPCN-333 may be more likely to diffuse away from C@FNPCN-333 before detoxification can occur. The ROS that escapes the nanofactories may then reach the cytosolic area and cause oxidative damage, inducing increased toxicity, as observed.

In contrast to the proximity effect, altering the localization of SC@FNPCN-333 (endocytic vs. cytosolic) did not result in significant difference in protecting cells from PQ-induced oxidative stress. On one hand, this is surprising as one may expect cytosolic SC@FNPCN-333 to come into contact with more diffusing ROS than SC@FNPCN-333 trapped inside endocytic

organelles. On the other hand, it is interesting to note that endosomes redox active and that they contain proteins capable of mediating the transport of ROS across bilayers[68–70]. It is therefore possible that a significant portion of PQ-generated superoxide reaches the lumen of endocytic organelles. Under such scenario, endocytic and cytosolic SC@FNPCN-333 would show similar activities by detoxifying cells from the damaging effects of superoxide, before or after in penetrates endosomes, respectively.

Given the multiple criteria that have to be taken into account to generate optimal nanofactories, it is difficult to predict which enzymes may be compatible with MOF encapsulation. Several aspects of a MOF-based strategy however point to a potentially broad applicability. For instance, the diameter of the MOF cavities is tunable, providing opportunities for the encapsulation of enzymes of various sizes. It is also possible to encapsulate a cocktail of enzymes, something that can lead to synergistic effects as demonstrated by the fact that SC@FNPCN-333 clearly outperforms the combination of S@FNPCN-333 and C@FNPCN-333. Additionally, the size and surface of the MOF can also be modified. This in turn could facilitate development of nanofactories that reside inside cells for even longer periods of time. Moreover, the compatibility of MOF nanofactories with dfTAT mediated cytosolic delivery highlights that these materials have access to the cytosolic space of live cells. While it did not lead to improvements in the context of SC@FNPCN-333, this is presumably important for the future development of nanofactories that would involve substrates that are confined within the cytosolic space, as is the case for numerous molecules associated with different metabolic pathways.

A side by side comparison between the nanofactories presented in this study and other reported enzyme protection techniques is difficult[71–73]. This is in part because the enzymes encapsulated often vary, persistence in cells is often not documented, and because each technique has a different set of advantages and disadvantages. Nonetheless, we propose that in vitro performances of the nanofactories achieved in this proof-of-concept study are extremely encouraging. Moreover, MOFs provide specific benefits over other encapsulation materials. For instance, liposomes, the most common enzyme carrier system, protect enzymes from degradation effectively. However, the lipid bilayer of liposomes forms a barrier between enzyme and substrate. Enzyme protection therefore comes at the cost of a severe reduction in enzymatic activity, unless complex strategies are implemented to permit substrate diffusion or controlled enzyme release. In contrast, MOF-encapsulated enzymes remain accessible to substrate without the need of enzyme release from the carrier. As a matter of fact, the activity of encapsulated enzymes is comparable to that of the enzymes in their free form. MOF encapsulation, and protection from proteolytic degradation, therefore does not compromise enzymatic activity. This is turn allows for simple design and nanparticle synthesis (i.e. no need for release strategies).

While the biocompatibility of MOF, in cellulo and in vivo, certainly needs to be further tested, these results point to various potential biotechnological approaches. For instance, by varying the enzymes encapsulated within the MOF structures, nanofactories such as those described herein may replace defective metabolic activities and contribute to enzyme replacement therapies. Moreover, we envision that MOF-enzyme nanofactories may find applications in cell biology and ex vivo cell engineering. For instance, SC@FNPCN-333 may be readily useful for cultures of cells prone to oxidative stress (e.g., primary cells). MOF-enzyme nanofactories may also be useful as organelle mimetics that confer cells novel properties, including resistance to various stresses[74–78]. We also envision that MOF-enzyme nanofactories, if combined with reporter assays, could allow the

probing of metabolic activities in cell cultures over extended periods of time.

## Methods

**Synthesis of NPCN-333**. 10 mL DMF solution of AlCl$_3 \cdot$ 6H$_2$O (1.5 mg mL$^{-1}$), 5 mL DMF solution of TATB (1 mg mL$^{-1}$), 15 mL DMF and 50 μL TFA was mixed and heated at 95 °C for 24 h. NPCN-333 was collected by centrifugation.

**Synthesis of FNPCN-333**. 30 mg NPCN-333 was dispersed in 5 mL DMF in which was added 5 mL 10 mg mL$^{-1}$ DMF solution of BTB-Green. The mixture was kept in 85 °C oven for 4 h and the solid was collected by centrifugation. The determination of the amount of metathesized ligand was conducted by digesting the obtained solid in HCl, dried under vacuum and dissolved in deuterated DMSO for NMR analysis. The ligand ratio of BTB-Green/TATB is 1:6.

**Stepwise encapsulation of SOD and CAT on FNPCN-333**. Stock solutions of SOD (5 mg mL$^{-1}$) and CAT (5 mg mL$^{-1}$) were prepared by dissolving SOD and CAT in deionized water, respectively. 1 mg FNPCN-333 was suspended in water in which 1 mL CAT stock solution was added. The mixture was vortexed for 20 min and the solid was collected by centrifugation and washed by fresh water for 3 times. The solid was re-suspended in 1 mL water and 1 mL SOD stock solution was added. The mixture was kept vortexing for 20 min and the solid was collected by centrifugation. SC@FNPCN-333 was washed with fresh water for 3 times before re-suspended in 0.3 mL water.

**WST assay for determining SOD activity**. WST assay kit is purchased from Sigma Aldrich. 20 μL sample solution containing different concentrations of SC@FNPCN-333 is mixed with 200 μL working solution. Then 20 μL WST solution (1 mL WST stock solution diluted by 19 mL working solution) is added and well mixed. Finally 20 μL xanthine oxidase (XOD) solution (15 μL XOD stock solution diluted with 2.5 mL working solution) is added and the solution is incubated at 37 °C for 30 min. The reading at 450 nm is collected by UV-vis spectroscopy.

**Amplex Red-HRP assay for determining CAT activity**. 20 μL sample solution containing different concentrations of SC@FNPCN-333 is mixed with 200 μL hydrogen peroxide PBS solution (final concentration is 40 μM) and incubates at 37 °C for 30 min. Then 20 μL HRP solution (0.4 U mL$^{-1}$) and 20 μL Amplex Red solution (100 μM) is added and incubates at 37 °C for another 30 min. Fluorescence is collected with excitation wavelength of 540 nm and emission wavelength of 590 nm.

**Cell internalization of SC@FNPCN-333 and CLSM imaging**. HeLa cells were seeded in an 8 well plate and allowed to adhere overnight. Then the culture media was replaced by 200 μL fresh nrL-15 media containing SC@FNPCN-333 with different concentrations at 37 °C for 15–480 min in darkness. The cells used for CLSM imaging was cultured in 75 μg mL$^{-1}$ SC@FNPCN-333 for 2 h. Representatively, for the CLSM imaging, media containing SC@FNPCN-333 was removed and cells were washed with fresh nrL-15 for 3 times before they were stained with Hoechst 33,342 (5 μg mL$^{-1}$), Lyso Tracker red (10 μg mL$^{-1}$) and SYTOX Blue (5 μg mL$^{-1}$). The cells were then incubated for 5 min before they were imaged. Co-incubation of SC@FNPCN-333 and D-dfTAT (20 μM) was conducted in the same manner. The only difference lied in the cell staining step. The cells were only stained with SYTOX Blue.

**Cell lysis protocol**. Cells are cultured in a 48 well plate and treated before the culture medium is removed. Cells are washed with fresh PBS for three times and five representative images are obtained by a confocal microscopy to count the cell number in the plate. Then the cells are digested by 200 μL concentrated nitric acid overnight. Each sample is measured three times by ICP-MS.

**Intracellular Al content measurement at day 7**. Cells (80–90% confluency) is cultured with 75 μg mL$^{-1}$ SC@FNPCN-333 in nrL-15 for 2 h. Then the cells are washed with frest nrL-15 for three times and cultured with fresh DMEM. Cells are trypsinized and re-plated at day 1, 3, and 5. At day 7 the culture medium is removed and the cells are treatment with concentrated nitric acid overnight. The sample is measured three times by ICP-MS. Original Al content value: 104.1 ± 2.8 ng Al/10$^5$ cells, mean ± s.d.

**In vitro antioxidative activity evaluation of SC@FNPCN-333**. HeLa cells were seeded in a 48 well plate and allowed to adhere overnight. For the positive control groups, cell culture media was replaced by 200 μL fresh DMEM media containing PQ with concentrations from 0.5 mM to 10 mM. The cells were cultured at 37 °C for 24 h, then the cells were washed with fresh PBS buffer for three times before charged with fresh nrL-15 buffer. The cells were stained with Hoechst 33,342 (5 μg mL$^{-1}$) and SYTOX Green (5 μg mL$^{-1}$). To evaluate the antioxidative activity of SC@FNPCN-333, cells were treated with 75 μg mL$^{-1}$ SC@FNPCN-333

for 2 h before charged with PQ solutions for 24 h. The washing and staining operations were the same as positive controls. Hoechst 33,342 was excited for 500 ms and SYTOX Green was excited for 300 ms. 5 images were taken in each well. Each treatment condition was replicated for 3 wells.

**Real-time ROS monitoring in living cells**. Cells are cultured in a 48 well plate and are treated with SC@FNPCN-333 for 2 h in nrL-15. Then the medium is removed and the cells are washed with fresh nrL-15 for three times. Then the cells are incubated with nrL-15 containing 500 μM pyocianin and 5 μM superoxide detection dye for 30 min before imaged by a confocal microscope.

**Theoretical estimation of enzyme loading in NPCN-333**. In each unit cell of PCN-333, there are eight of A-cages (5.5 nm) and 16 of B-cages (4.2 nm). The volume of each unit cell = (126 Å)$^3$ = $2.0 \times 10^{-18}$ cm$^3$. The density of PCN-333(Al) = 0.23 g/cm$^3$. So the mass of each unit cell = $\rho \times V = 0.46 \times 10^{-18}$ g. Therefore, the total number of unit cells per gram of PCN-333(Al) is: $1/(0.46 \times 10^{-18}) = 2.2 \times 10^{18}$. And the A-cage in each gram of PCN-333(Al) = $2.2 \times 10^{18} \times 8 = 1.7 \times 10^{19}$ = $2.9 \times 10^{-5}$ mol. B-cage in each gram of PCN-333(Al) = $3.4 \times 10^{19} = 5.8 \times 10^{-5}$ mol. For SOD, $M_W$ = 16.3 kDa, so the maximum loading is $16,300 \times 5.8 \times 10^{-5}$ = 0.92 g g$^{-1}$. For CAT, $M_W$ = 64 kDa, so the maximum loading is $60,000 \times 2.9 \times 10^{-5}$ = 1.74 g g$^{-1}$.

**Theoretical estimation of enzyme loading in FNPCN-333**. In each unit cell of PCN-333, there are eight of A-cages (5.5 nm) and 16 of B-cages (4.2 nm). The volume of each unit cell = (126 Å)$^3$ = $2.0 \times 10^{-18}$ cm$^3$. The density of FNPCN-333(Al) = 0.26 g/cm$^3$. So the mass of each unit cell = $\rho \times V = 0.52 \times 10^{-18}$ g. Therefore, the total number of unit cells per gram of PCN-333(Al) is: $1/(0.52 \times 10^{-18}) = 1.95 \times 10^{18}$. And the A-cage in each gram of PCN-333(Al) = $1.95 \times 10^{18} \times 8 = 1.5 \times 10^{19} = 2.6 \times 10^{-5}$ mol. B-cage in each gram of PCN-333(Al) = $3.0 \times 10^{19} = 5.2 \times 10^{-5}$ mol. For SOD, $M_W$ = 16.3 kDa, so the maximum loading is $16,300 \times 5.2 \times 10^{-5} = 0.82$ g g$^{-1}$. For CAT, $M_W$ = 64 kDa, so the maximum loading is $60,000 \times 2.6 \times 10^{-5} = 1.56$ g g$^{-1}$.

**Co-localization coefficient estimation**. For cellular co-localization experiments, the Manders' overlap coefficient $R$ (measures how interdependent the red and green channels are) and co-localization coefficient M1 (measures the percentage of above-background pixels in the red channel that overlap with the above-background pixels in the green channel) were calculated using ImageJ (NIH).

**Data availability**. All data are available from the authors upon reasonable request.

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

## Acknowledgements

This article was supported by the Welch Foundation through a Robert A. Welch Chair in Chemistry to H.C.Z. (A-0030) and by a grant from the National Institutes of Health (to J.P.P., award number R01GM110137).The TEM acquisition was supported by the NSF grant DBI-0116835, the VP for Research Office, and the TX Eng. Exp. Station. The authors thank Dr. Hansoo Kim for acquiring TEM images, Dr. Yu Fang, Mr. Jialuo Li, Dr. Kristina Najjar, Mr. Jason Allen, Ms. Helena Kondow, Mr. Dakota Brock, Dr. Ting-Yi Wang, Dr. Xuan Wang, Mr. Zhiyuan Jiang and Dr. Lu-Jia Liu for experimental help and fruitful discussion

## Author contributions

X.L., J.-P.P., and H.-C.Z. conceived of the project. X.L. and A.E.-O. carried out the experimental work. X.L., A.E.-O., J.-P.P., and H.-C.Z. analyzed the data. X.L., J.-P.P., and H.-C.Z. wrote the manuscript.

## Additional information

**Competing interests:** The authors declare no competing financial interests.

