## [Peer Review File · Nature Communications]

Reviewer #1 (Remarks to the Author):

This is a very nice piece of work using MOF as nanocarriers for the highly efficient intracellular delivery of two therapeutic enzymes. First of all, to the best of my knowledge, this is the first example of using MOF for protein delivery. Enzyme immobilization on MOFs has generated much interest in recent years, due to the high protection effect of MOFs for protein structure and thus the highly improved stability for enzymes. From some up-to-date review articles (Catalysis Science & Technology 2015, 5, 5077-5085; Chemical Society Reviews 2017, 46, 3386-3401), we can see that the incorporation of enzyme in MOFs has demonstrated its great capability of increasing enzyme stability at harsh conditions while preserving enzyme activity at the same time for biocatalysis applications. By this study, Zhou and coworkers, for the first time, demonstrated that the well-designed MOF scaffold can be utilized for highly efficient delivery of enzyme into cells. This study opens many possibilities for MOFs in biomedical applications, especially in protein therapy.

The study has three THRUSTS:

THRUST 1: Two enzymes, SOD and CAT, with different dimensions, were colocalized into respective pores of a well-designed MOF having mesopores of two different sizes.

For this thrust, the authors synthesized a special MOF for colocalization of two enzymes and carefully characterized the structure of the composite, demonstrating the successful incorporation of two enzymes with high enzyme loadings.

Question: The high enzyme loading was determined by BCA method in this study, and I am wondering whether TGA analysis could be used to further confirm the high loading of protein.

THRUST 2: SOD and CAT incorporated in MOF displayed very high stability in the presence of acid pH, protease, etc, which are essential for protein delivery into cells. At the same time, the incorporated SOD and CAT both showed highly retained (~100%) activities compared to free enzymes.

For this thrust, the authors carried out a detailed study to prove the high stability and high activity of incorporated enzymes at various harsh conditions. The control experiments in this section are appropriate and complete and I am convinced by the claims and conclusions.

THRUST 3: SOD and CAT incorporated in MOF were delivered into cells and worked efficiently in vitro in cells.

For this thrust, many cell experiments were designed to demonstrate the localization and high uptake of the composite in cells. More importantly, the delivered enzymes can significantly reduce the ROS-induced toxicity for cells and improve cell survival obviously, while free enzymes cannot do so. In addition, the authors also showed that the MOF carrier has non-toxicity to cells.

Question: In Figure 4b, the result showed that SC@FNPCN-333 worked much better than S@FNPCN-333 + C@FNPCN-333. The authors proposed that it might be the proximity effect of two enzymes in MOF caused the improved overall anti-oxidant activity of the bi-enzyme system. The proximity effect usually works in enzyme cascade reactions; here in this study is the bi-enzyme reaction also a cascade reaction? Or, there is some synergistic effect between the two enzymatic reactions? More discussion needs to be provided.

Overall, this is a very nice study with exciting results, showing previously unknown and new possibilities for the application of MOFs. The manuscript was written in a well-organized manner as well. I suggest its publication after addressing above minor points.

Reviewer #2 (Remarks to the Author):

The manuscript describes the preparation of MOF materials that are then utilised as supports for two enzymes superoxide dismutase and catalase. The novelty of this work lies in the use of the immobilised enzymes to assist as anti-oxidants within cells. The manuscript details the preparation and characterisation of the materials. The immobilisation of the enzymes occurs via docking into external pores of the MOF. The catalytic activity of the two enzymes is comparable to the activity of the free enzymes and is active for a period of up to 7 days. The enzymes are stable in the presence of trypsin indicating that the MOF protects the enzymes from proteolytic degradation. Cellular uptake of the enzyme modified MOF was demonstrated by using AAS to monitor the Al content of the cells, while the anti-oxidant capacity of the MOF system was ascertained by challenging cells with paraquat. The ability of the MOF system to protect cells was evident over a period of 7 days.

This work is very detailed, it shows a novel use of MOF materials as enzyme supports within cells, in my opinion it is suitable for publication.

Two minor comments:

line 132 hydrogen hydroxide should be hydrogen peroxide

one of the authors inserted a comment on the figure labels which needs to be removed, I agree fully with the sentiments expressed in the comment.

Reviewer #3 (Remarks to the Author):

Lian's paper deals with the encapsulation of two enzymes in a nanometric metal organic framework and the in vitro evaluation of their antioxidant activity.

The manuscript presents some nice results. The validity of the approach and the quality of data are consistent. However, the lack of novelty does not allow their publication in a prestigious journal such as Nature Communications. This paper would be suitable for other more specific journals. From the viewpoint of enzyme-associated MOFs and nanometric MOFs, the novelty of this study is limited, as such composites have been subject of investigation in several works and reviews (see: T.J. Piklak et al., *Top Catal.* 2006; V. Lykourinou et al., *JACS*, 2011 ; Y. Chen et al, *Inorg Chem*, 2012; W.L. Liu et al., *Chem.*, 2014 ; D. Feng et al., *Nat. Comm.* 2015 ; P. Li et al., *ACS Nano* 2016; E. Gkaniatsou et al., *Mater Horiz.*, 2017; X. Lian et al, *Chem. Soc. Rev.* 2017, among others). Further, the association of enzymes to the PCN-333 material has already been reported by some of these authors (*Nat. Comm.*, 2015, 6, 5979). On the other site, in vitro antioxidant activity of some MOFs has been previously reported (L. Cooper et al., *Chem. Comm.* 2015; T. Hidalgo et al., *J. Mater Chem B*, 2017)

Despite the important amount of results, there is no enough discussion. Discussion section seems more a conclusion part. Authors are encouraged to add more discussion.

The following is a list of my specific comments to this manuscript:

- As mentioned by authors in the introduction section, proteins may reach the location where activity is required. However, this is not only an issue at the cellular lever, but at the organism level. Therefore, biodistribution and potential targeting of these nanoparticles needs to be addressed to claim the enzyme replacement therapies in humans.

- Encapsulation of enzymes in order to improve their transport properties, protein half-lives and stability has been widely studied using different nanocarriers (liposomes, silica, etc). A clear comparison between the performances of the proposed system and some of the best reported carriers should be included.

- In contrast with the affirmation "the compatibility of MOFs with cells has thus far not been investigated". Many in vitro tests (H. Su et al, *Chem. Comm.*, 2015; I.B. Vasconcelos et al., *RSC Acv.* 2012; X. Zhu et al., *Chem. Comm.*, 2014; Q. Hu et al. *J. Med. Chem*, 2014; F.R.S. Lucena et al., *Biomed. Pharamcother.* 2013) and some in vivo studies (T. Baati et al, *Chem. Sci.* 2013; T. Kundu et

al., Chem. Eur. J. 2014 ; A. Ruyra et al., Chem. Eur. J., 2015) dealing with the toxicity of some MOFs have already been reported. In addition, the presence of Al within the proposed MOF, as well as remaining DMF from the synthesis, could be a limitation for the in vivo use. Please, address this point in the manuscript.

- Biological properties (e.g. cell internalization) of the NPCN-333 and FNPCN-333 could be different due to the presence of the fluorophore, as the interaction with the biological surrounding could be modified.
 - After 7 days in contact with cell culture media, PXRD of FNPCN-333 exhibits a peak broadening due to a probable partial degradation. Degradation of the nanoparticles might be quantitatively addressed by determining the amount of ligand and Al released to the intracellular and extracellular media.
 - Isotherms and pore size distribution of the FNPCN-333 vs. PCN-333 and vs. C@/SC@ FNPCN-333 might be represented overlapped for a better comparison. A priori, the presence of the fluorophore moiety might lead to a decrease in the mesoporosity and not microporosity. However, it seems that microporosity is also affected.
 - Figure S14 shows larger NPCN-333 particles after 7 days-soaking in DMEM. Please, comment on that.
 - Please, note if the error in Table S1 corresponds to SD or SEM. Note that SD would be here more appropriated. Particle size and surface charge might be determined in cell culture media and other relevant biological fluids.
 - Why HeLa cells have been selected as in vitro model?
 - Viability tests of SC@ FNPCN-333 in absence of PQ (HeLa cells without treatment) might be included at longer times (7 d).
 - Further discussion should be included to explain similar activity of SC@ FNPCN-333 in presence and absence of dFTAT (cytosolic vs. endocytic location).
 - As finally the nanoparticles location in the cell has not effect on the activity, what is the relative SOD and CAT activity after 24 h at pH 7.4 (cytosol)?
 - The Figure S19 does not really support the affirmation that the green fluorescence (SC@ FNPCN-333 endocytosis) is greatly diminished at 4°C vs. 37°C.
 - In vitro antioxidant tests (in presence of PQ and pyocyanin) should be compared with similar studies reported for other promising nanoparticles within the field.
 - Figure 3c and d might be also expressed in % of Al considering the initial Al dose in order to estimate the percentage of internalized nanoparticles.
- SC@ FNPCN-333 should be characterized after 7 days in contact with cells (PQ studies), using confocal microscopy and ICP methods.
- Enzyme leaching should be also studied at pH 7.4 and compare with the data at pH 5.

- Please, refer to the supporting information for the calculation of the maximal encapsulation capacity. Similar simple estimation could be performed after the material methatesis.

- Specific experimental protocols (media, concentration, cell number, time, etc) are needed for WDST enzyme activity, cell culture, staining procedure, cell lysis protocol for ICP determination, dfTAT studies, pyocianin studies, etc.

Reviewers' comments:

Reviewer #1 (Remarks to the Author):

This is a very nice piece of work using MOF as nanocarriers for the highly efficient intracellular delivery of two therapeutic enzymes. First of all, to the best of my knowledge, this is the first example of using MOF for protein delivery. Enzyme immobilization on MOFs has generated much interest in recent years, due to the high protection effect of MOFs for protein structure and thus the highly improved stability for enzymes. From some up-to-date review articles (Catalysis Science & Technology 2015, 5, 5077-5085; Chemical Society Reviews 2017, 46, 3386-3401), we can see that the incorporation of enzyme in MOFs has demonstrated its great capability of increasing enzyme stability at harsh conditions while preserving enzyme activity at the same time for biocatalysis applications. By this study, Zhou and coworkers, for the first time, demonstrated that the well-designed MOF scaffold can be utilized for highly efficient delivery of enzyme into cells. This study opens many possibilities for MOFs in biomedical applications, especially in protein therapy.

The study has three THRUSTS:

THRUST 1: Two enzymes, SOD and CAT, with different dimensions, were colocalized into respective pores of a well-designed MOF having mesopores of two different sizes. For this thrust, the authors synthesized a special MOF for colocalization of two enzymes and carefully characterized the structure of the composite, demonstrating the successful incorporation of two enzymes with high enzyme loadings.

Question: The high enzyme loading was determined by BCA method in this study, and I am wondering whether TGA analysis could be used to further confirm the high loading of protein.

The reviewer is correct in pointing out that TGA analysis would be useful to further confirm the high loading of protein. We included the results in supporting information (Figure S16-18) and discussion in the text:

“Thermal gravimetric analysis (TGA) further confirms high enzyme loading of FNPCN-333 (Figure S17).”

THRUST 2: SOD and CAT incorporated in MOF displayed very high stability in the presence of acid pH, protease, etc, which are essential for protein delivery into cells. At the same time, the incorporated SOD and CAT both showed highly retained (~100%) activities compared to free enzymes. For this thrust, the authors carried out a detailed study to prove the high stability and high activity of incorporated enzymes at various

harsh conditions. The control experiments in this section are appropriate and complete and I am convinced by the claims and conclusions.

THRUST 3: SOD and CAT incorporated in MOF were delivered into cells and worked efficiently in vitro in cells. For this thrust, many cell experiments were designed to demonstrate the localization and high uptake of the composite in cells. More importantly, the delivered enzymes can significantly reduce the ROS-induced toxicity for cells and improve cell survival obviously, while free enzymes cannot do so. In addition, the authors also showed that the MOF carrier has non-toxicity to cells.

Question: In Figure 4b, the result showed that SC@FNPCN-333 worked much better than S@FNPCN-333 + C@FNPCN-333. The authors proposed that it might be the proximity effect of two enzymes in MOF caused the improved overall anti-oxidant activity of the bi-enzyme system. The proximity effect usually works in enzyme cascade reactions; here in this study is the bi-enzyme reaction also a cascade reaction? Or, there is some synergistic effect between the two enzymatic reactions? More discussion needs to be provided.

To address the reviewer's comment, new text was added to the discussion section. In particular:

“It is notable that the proximity of SOD and CAT in the enzymatic nanoreactors enhances the protective effect against PQ induced oxidative stress. SOD catalyzes the disproportionation of superoxide and generates H_2O_2 and oxygen, while H_2O_2 is consumed by CAT, yielding water and oxygen. Since the two reactions catalyzed by SOD and CAT may occur in a cascade manner, it is likely that SOD-generated H_2O_2 is degraded by CAT before this ROS diffuses away from the nanoreactor. Conversely, in the case of MOF nanoparticles carrying only one enzyme, the H_2O_2 generated by S@FNPCN-333 may be more likely to diffuse away from C@FNPCN-333 before detoxification can occur. The ROS that escapes the nanofactories may then reach the cytosolic area and cause oxidative damage, inducing increased toxicity, as observed.”

Overall, this is a very nice study with exciting results, showing previously unknown and new possibilities for the application of MOFs. The manuscript was written in a well-organized manner as well. I suggest its publication after addressing above minor points.

Reviewer #2 (Remarks to the Author):

The manuscript describes the preparation of MOF materials that are then utilised as supports for two enzymes superoxide dismutase and catalase. The novelty of this work lies in the use of the immobilised enzymes to assist as anti-oxidants within cells. The manuscript details the preparation and characterisation of the materials. The immobilisation of the enzymes occurs via docking into external pores of the MOF. The catalytic activity of the two enzymes is comparable to the activity of the free enzymes and is active for a period of up to 7 days. The enzymes are stable in the presence of trypsin indicating that the MOF protects the enzymes from proteolytic degradation. Cellular uptake of the enzyme modified MOF was demonstrated by using AAS to monitor the Al content of the cells, while the anti-oxidant capacity of the MOF system was ascertained by challenging cells with paraquat. The ability of the MOF system to protect cells was evident over a period of 7 days.

This work is very detailed, it shows a novel use of MOF materials as enzyme supports within cells, in my opinion it is suitable for publication.

Two minor comments:

line 132 hydrogen hydroxide should be hydrogen peroxide

one of the authors inserted a comment on the figure labels which needs to be removed, I agree fully with the sentiments expressed in the comment.

The issues raised by the reviewer have been corrected in the text.

Reviewer #3 (Remarks to the Author):

Lian's paper deals with the encapsulation of two enzymes in a nanometric metal organic framework and the in vitro evaluation of their antioxidant activity.

The manuscript presents some nice results. The validity of the approach and the quality of data are consistent. However, the lack of novelty does not allow their publication in a prestigious journal such as Nature Communications. This paper would be suitable for other more specific journals. From the viewpoint of enzyme-associated MOFs and nanometric MOFs, the novelty of this study is limited, as such composites have been subject of investigation in several works and reviews (see: T.J. Piklak et al., Top Catal. 2006; V. Lykourinou et al., JACS, 2011 ; Y. Chen et al, Inorg Chem, 2012; W.L. Liu et al., Chem., 2014 ; D. Feng et al., Nat. Comm. 2015 ; P. Li et al., ACS Nano 2016; E. Gkaniatsou et al., Mater Horiz., 2017; X. Lian et al, Chem. Soc. Rev. 2017, among others). Further, the association of enzymes to the PCN-333 material has already been reported by some of these authors (Nat. Comm., 2015, 6, 5979). On the other site, in

vitro antioxidant activity of some MOFs has been previously reported (L. Cooper et al., Chem. Comm. 2015; T. Hidalgo et al., J. Mater Chem B, 2017)

Despite the important amount of results, there is not enough discussion. Discussion section seems more a conclusion part. Authors are encouraged to add more discussion.

The reviewer raises a concern about the novelty of our work. In addition, he encourages us to expand our discussion section. We have done so by addition of multiple important paragraphs (described below). These paragraphs put our work in better context, highlighting more effectively what has been done before (including all the citations provided by the reviewer), what the value of our work is, and how our work fits in the context of future applications. Overall, this expanded discussion, along with a modified introduction, should illustrate the novelty of our work more clearly and more convincingly.

The following is a list of my specific comments to this manuscript:

- As mentioned by authors in the introduction section, proteins may reach the location where activity is required. However, this is not only an issue at the cellular level, but at the organism level. Therefore, biodistribution and potential targeting of these nanoparticles needs to be addressed to claim the enzyme replacement therapies in humans.

We agree with the reviewer that our current data do not support a claim about the immediate usefulness of the protein-MOF system presented for enzyme replacement therapies in human. We do not make such claim. Instead we recognize that it is a first step towards such an ambitious goal. Notably, establishing biodistribution and targeting of these particles, would not be sufficient to establish the claim raised by the reviewer (they would only constitute an additional step toward that claim).

To address the reviewer's comment, new text was added in our discussion. This text addresses both the novelty of our work (as mentioned above by the reviewer as a concern) and redefines more clearly what our conclusions are. In particular, we emphasize with great care that, while enzyme replacement therapy is a potential long-term goal, other applications with more immediate feasibility are also possible.

“While the biocompatibility of MOF, *in cellulo* and *in vivo*, certainly needs to be further tested, these results point to various potential biotechnological approaches. For instance, by varying the enzymes encapsulated within the MOF structures, nanofactories such as those described herein may replace defective metabolic activities and contribute to enzyme replacement therapies. Moreover, we envision that MOF-enzyme nanoreactors may find applications in cell biology and *ex vivo* cell engineering. For instance, SC@FNPCN-333 may be readily useful for cultures of cells prone to oxidative stress (e.g. primary cells). MOF-enzyme nanoreactors may also be useful as organelle mimetics that confer cells novel properties, including resistance to various stresses.(REF) We also envision that MOF-enzyme nanoreactors, if combined with reporter assays, could allow the probing of metabolic activities in cell cultures over extended periods of time. ”

Encapsulation of enzymes in order to improve their transport properties, protein half-lives and stability has been widely studied using different nanocarriers (liposomes, silica, etc). A clear comparison between the performances of the proposed system and some of the best reported carriers should be included.

On one hand, the reviewer is correct in pointing out that a comparison between different nanocarriers would, in principle, be informative. On the other hand, we believe that establishing a “clear comparison” is not possible and simply inadequate. In particular, the efficacy of various nanocarriers depends on many parameters: in this context comparing MOF to liposomes or other systems is equivalent to comparing apples to oranges.

For instance, liposomes have known issues in terms of substrate accessibility. It is potentially relevant in the case of the enzyme system described herein as superoxide may not cross the lipid bilayer of a liposome readily. Overall, the performance of the liposomes encapsulation may therefore lead to poor results in this case. Yet, the same liposomes could perform well for other enzymes that process membrane-permeable substrates. Then, would a comparison be clear, and more importantly, fair?

Nonetheless, to address the reviewers’ comment, we have added text in our discussion section. This new text highlights the benefits of MOFs and those of other carrier systems. It also describes how these nanocarrier systems have

different pros and cons. We believe that readers will now have a better appreciation for the value of each approach.

“A side by side comparison between the nanofactories presented in this study and other reported enzyme protection techniques is difficult.⁷¹⁻⁷³ This is in part because the enzymes encapsulated often vary, persistence in cells is often not documented, and because each technique has a different set of advantages and disadvantages. Nonetheless, we propose that *in vitro* performances of the nanofactories achieved in this proof-of-concept study are extremely encouraging. Moreover, MOF provide specific benefits over other encapsulation materials. For instance, liposomes, the most common enzyme carrier system, protect enzymes from degradation effectively. However, the lipid bilayer of liposomes forms a barrier between enzyme and substrate. Enzyme protection therefore comes at the cost of a severe reduction in enzymatic activity, unless complex strategies are implemented to permit substrate diffusion or controlled enzyme release. In contrast, MOF-encapsulated enzymes remain accessible to substrate without the need of enzyme release from the carrier. As a matter of fact, the activity of encapsulated enzymes is comparable to that of the enzymes in their free form. MOF encapsulation, and protection from proteolytic degradation, therefore does not compromise enzymatic activity. This in turn allows for simple design and nanoparticle synthesis (i.e. no need for release strategies).”

- In contrast with the affirmation “the compatibility of MOFs with cells has thus far not been investigated”. Many *in vitro* tests (H. Su et al, Chem. Comm., 2015; I.B. Vasconcelos et al., RSC Adv. 2012; X. Zhu et al., Chem. Comm., 2014; Q. Hu et al. J. Med. Chem, 2014; F.R.S. Lucena et al., Biomed. Pharmacother. 2013) and some *in vivo* studies (T. Baati et al, Chem. Sci. 2013; T. Kundu et al., Chem. Eur. J. 2014 ; A. Ruyra et al., Chem. Eur. J., 2015) dealing with the toxicity of some MOFs have already been reported. In addition, the presence of Al within the proposed MOF, as well as remaining DMF from the synthesis, could be a limitation for the *in vivo* use. Please, address this point in the manuscript.

The reviewer is correct in pointing out that our original assertion “the compatibility of MOFs with cells has thus far not been investigated” is inaccurate. Our phrasing was gauche and our intention was not to ignore the work of others. To address the reviewer’s comment, we have change our introduction

Although the biocompatibility of some MOF materials have been investigated in a number of reports, whether MOF-enzyme composites may serve as efficient nanoreactors in living cells remains untested.

We have changed the phrase in the manuscript. For the possible residue of DMF, because MOF nanoparticles have been extensively washed with water before enzymes are introduced, and since another round of washing is conducted after enzymes are incubated with MOF particles, the DMF should have been completely removed. The presence of Al in the material seems to be non-deleterious to cells. According to one of our recent studies the same MOF does not cause weight loss in mice, which suggests the nontoxic nature of the MOF in vivo.

Biological properties (e.g. cell internalization) of the NPCN-333 and FNPCN-333 could be different due to the presence of the fluorophore, as the interaction with the biological surrounding could be modified.

Answer: The reviewer's point is correct. Owing to this reason, we used FNPCN-333 for all of the in cellulo studies.

- After 7 days in contact with cell culture media, PXRD of FNPCN-333 exhibits a peak broadening due to a probable partial degradation. Degradation of the nanoparticles might be quantitatively addressed by determining the amount of ligand and Al released to the intracellular and extracellular media.

Figure S14 shows larger NPCN-333 particles after 7 days-soaking in DMEM. Please, comment on that.

The reviewer is correct in pointing out the peak broadening observed in Figure 1. While we agree that this is indicative of partial degradation of the material, the TEM data in Figure S14 demonstrate that crystallinity of the material is nonetheless partially conserved. In particular, we showed zoom-in images in Figure S11, demonstrating “the fringes on the crystals”. Notably, based on DLS results the particle size does not significantly change before and after soaking in DMEM for 7 days. The swelling of particles in TEM images may be ascribed to the unexpected aggregation during the preparation process of TEM specimen.

- Isotherms and pore size distribution of the FNPCN-333 vs. PCN-333 and vs. C@/SC@ FNPCN-333 might be represented overlapped for a better comparison. A priori, the presence of the fluorophore moiety might lead to a decrease in the

mesoporosity and not microporosity. However, it seems that microporosity is also affected.

The distribution of microporosity, that is, the peak area of microporosity over that of mesoporosity, of FNPCN-333 is more significant than that of NPCN-333. So the inclusion of a larger ligand does decrease mesoporosity. This is demonstrated in supporting information Figure S9.

This is now addressed in the text with “The presence of BTB-Green on the framework backbone is also confirmed by the relatively larger distribution of microporosity in FNPCN-333 than that of NPCN-333 (Figure S9).”

Please, note if the error in Table S1 corresponds to SD or SEM. Note that SD would be here more appropriated. Particle size and surface charge might be determined in cell culture media and other relevant biological fluids.

The error is SD. We have added it to the table caption. The particle size and surface charge is measured in DMEM buffer without FBS, because proteins may cause aggregation during the measurement.

Why HeLa cells have been selected as in vitro model?

HeLa cells are common vitro model for oxidative stress studies and this is the basis for our choice. We now describe this choice in our material and methods section. Our new text is also accompanied by citations that refer to HeLa cells as suitable models.

In addition, please note that we also tested the internalization and the antioxidative activity of the nanoparticles in mouse cells and human primary cells. The results are included in the supporting information (Figure S36-39).

Viability tests of SC@ FNPCN-333 in absence of PQ (HeLa cells without treatment) might be included at longer times (7 d).

The reviewer is correct in pointing out that this control is important. New data are now added to supporting information Figure S35. Our results show that SC@ FNPCN-333 does not reduce cell viability at 7 d.

Further discussion should be included to explain similar activity of SC@ FNPCN-333 in presence and absence of dfTAT (cytosolic vs. endocytic location).

New text in our discussion now describes the results obtained with SC@ FNPCN-333 in presence and absence of dfTAT.

“In contrast to the proximity effect, altering the localization of SC@FNPCN-333 (endocytic vs. cytosolic) did not result in significant difference in protecting cells from PQ induced oxidative stress. On one hand, this is surprising as one may expect cytosolic SC@FNPCN-333 to come into contact with more diffusing ROS than SC@FNPCN-333 trapped inside endocytic organelles. On the other hand, it is interesting to note that endosomes redox active and that they contain proteins capable of mediating the transport of ROS across bilayers.⁶⁸⁻⁷⁰ It is therefore possible that a significant portion of PQ-generated superoxide reaches the lumen of endocytic organelles. Under such scenario, endocytic and cytosolic SC@FNPCN-333 would show similar activities by detoxifying cells from the damaging effects of superoxide, before or after it penetrates endosomes, respectively.”

As finally the nanoparticles location in the cell has not effect on the activity, what is the relative SOD and CAT activity after 24 h at pH 7.4 (cytosol)?

SOD and CAT maintained about 90% of their original activities after 24 h at pH 7.4. This is described in Figure S17 and S18.

The Figure S19 does not really support the affirmation that the green fluorescence (SC@ FNPCN-333 endocytosis) is greatly diminished at 4°C vs. 37°C.

The reviewer is correct in noting that Figure S23 (formerly S19 in reviewed manuscript), in its original form, may not have been clear. To better address the issue of colocalization (or loss of colocalization), we have provided zoom-in images and added a colocalization analysis to measure this effect quantitatively. We modified our claim in the text “Consistent with these results, incubation of SC@FNPCN-333 with HeLa cells at 4 °C, a condition that inhibits energy-dependent endocytic uptake, or in the presence of amiloride, an inhibitor of macropinocytosis, led to a dramatic reduction in the number of intracellular SC@FNPCN-333-positive puncta and to a reduction in co-localization with LysoTracker red (Figure S23)”.

Listed in the image captions, a clear drop in correlation coefficients can be observed between the absence of amiloride at 37°C incubation (Figure 3, R=0.97, M1=0.99) and the presence of amiloride (Figure S23, R=0.34, M1=0.30) and 4°C incubation (R=0.61, M1=0.58). R corresponds to the Pearson’s coefficient, while M is the Mander’s coefficient. These two coefficients represent a quantitative measure of colocalization (as now indicated in Materials and Methods).

In vitro antioxidant tests (in presence of PQ and pyocyanin) should be compared with similar studies reported for other promising nanoparticles within the field.

The difficulties associated with such comparison were described above. To address the reviewer's comment, the following text was added to the discussion:

“A side by side comparison between the nanofactories presented in this study and other reported enzyme protection techniques is difficult.⁷¹⁻⁷³ This is in part because the enzymes encapsulated often vary, persistence in cells is often not documented, and because each technique has a different set of advantages and disadvantages. Nonetheless, we propose that *in vitro* performances of the nanofactories achieved in this proof-of-concept study are extremely encouraging. Moreover, MOF provide specific benefits over other encapsulation materials. For instance, liposomes, the most common enzyme carrier system, protect enzymes from degradation effectively. However, the lipid bilayer of liposomes forms a barrier between enzyme and substrate. Enzyme protection therefore comes at the cost of a severe reduction in enzymatic activity, unless complex strategies are implemented to permit substrate diffusion or controlled enzyme release. In contrast, MOF-encapsulated enzymes remain accessible to substrate without the need of enzyme release from the carrier. As a matter of fact, the activity of encapsulated enzymes is comparable to that of the enzymes in their free form. MOF encapsulation, and protection from proteolytic degradation, therefore does not compromise enzymatic activity. This in turn allows for simple design and nanoparticle synthesis (i.e. no need for release strategies).”

Figure 3c and d might be also expressed in % of AI considering the initial AI dose in order to estimate the percentage of internalized nanoparticles. SC@ FNPCN-333 should be characterized after 7 days in contact with cells (PQ studies), using confocal microscopy and ICP methods.

To address the reviewer's comment, % AI have been calculated and are now provided in Figure S26 and 27. In addition, titration experiments were performed to determine whether the protection activity at day 7 was consistent with the amount of AI detected.

These new results are added as Figures S43-45 and with the text below:

“To maximize their application and usefulness, enzyme nanofactories should have a persistent effect inside cells. To test whether SC@FNPCN-333 could sustain its protecting effect beyond 24 h, cells pre-incubated with SC@FNPCN-333 were cultured with fresh DMEM buffer for up to a week. PQ-induced toxicity was evaluated at day 2 and day 7. As shown in Figure 4d, treating cells with free SOD and CAT, which provided

only a weak protective effect at day 0, failed to rescue cells from PQ-induced cytotoxicity at day 1. In contrast, the rescuing effect of SC@NPCN-333 at day 2 was similar to that obtained at day 1. Moreover, this effect, although partially diminished, remained significant at day 7. This long-term persistence is consistent with the stability of the MOF material detected *in vitro*, where little protein leaching out of the MOF is observed over 7 days in an acidic milieu (Figure S40 and S41). Notably, ICP-MS analysis shows that the intracellular aluminum content present at day 7 was approximately 6% of that of cells at day 0 (1600 ng Al per 10^5 cells are detected after incubation of 75 $\mu\text{g}/\text{mL}$ of SC@FNPCN-333 at day 0, 104 ng Al per 10^5 cells at day 7; Figure S43). This is therefore indicative of a decline in overall MOF concentration of present in cells overtime, as would be expected from the dilution that take place during each cell division during one week of cell culture. However, titration experiments confirmed that lower doses of SC@FNPCN-333 could maintain high protection activities in cells. In particular, while incubation of cells at 1 $\mu\text{g}/\text{mL}$ SC@FNPCN-333 provides no protective effect, incubation with 2.5 $\mu\text{g}/\text{mL}$ SC@FNPCN-333 leads to approximately 100 ng Al per 10^5 cells and 60% cell viability after PQ treatment (a condition similar to that obtained at day 7 after incubation of 75 $\mu\text{g}/\text{mL}$ SC@FNPCN-333, Figures S44 and S45). Overall, these results indicate that the prolonged effect of SC@FNPCN-333 is imparted by the chemical stability of the material as well as its sustained enzymatic activity even at low intracellular concentrations.”

Enzyme leaching should be also studied at pH 7.4 and compare with the data at pH 5.

Accordingly to the reviewer’s request, the enzyme leaching at pH 7.4 is added to the supporting information Figure S41.

Please, refer to the supporting information for the calculation of the maximal encapsulation capacity. Similar simple estimation could be performed after the material metathesis.

The estimation method has been included in the materials and methods section.

“In each unit cell of PCN-333, there are eight of A-cages (5.5 nm) and 16 of B-cages (4.2 nm). The volume of each unit cell = $(126 \text{ \AA})^3 = 2.0 \times 10^{-18} \text{ cm}^3$. The density of FNPCN-333(Al) = $0.26 \text{ g}/\text{cm}^3$. So the mass of each unit cell = $\rho \times V = 0.52 \times 10^{-18} \text{ g}$. Therefore, the total number of unit cells per gram of PCN-333(Al) is: $1/(0.52 \times 10^{-18}) = 1.95 \times 10^{18}$. And the A-cage in each gram of PCN-333(Al) = $1.95 \times 10^{18} \times 8 = 1.5 \times 10^{19} = 2.6 \times 10^{-5} \text{ mol}$. B-cage in each gram of PCN-333(Al) = $3.0 \times 10^{19} = 5.2 \times 10^{-5} \text{ mol}$. For SOD, $M_w = 16.3 \text{ kDa}$, so the maximum loading is $16300 \times 5.2 \times 10^{-5} = 0.82 \text{ g}/\text{g}$. For CAT, $M_w = 64 \text{ kDa}$, so the maximum loading is $60000 \times 2.6 \times 10^{-5} = 1.56 \text{ g}/\text{g}$.”

Specific experimental protocols (media, concentration, cell number, time, etc) are needed for WDST enzyme activity, cell culture, staining procedure, cell lysis protocol for ICP determination, dfTAT studies, pyocyanin studies, etc.

All of the protocols requested are now included in the supporting information.

Reviewer #1 (Remarks to the Author):

The authors have carefully addressed the concerns raised by reviewers in the revised manuscript. Therefore I recommend its publication.

Reviewer #2 (Remarks to the Author):

The authors have provided a full and detailed response to the comments raised by the reviewers. Additional experimental data (TGA analysis and a range of fluorescence results) have been included. Discussion on the following points has been included:
the possibility of a protective effect due to the close proximity of SOD and CAT to reduce detrimental effects of peroxide

questions on the novelty of the work have been addressed by the inclusion of a discussion comparing the work presented with and other nanostructured enzymatic carriers such as liposomes that indicates the potential advantages of the MOF based system

potential detrimental effects of DMF and Al are shown not to arise

a rationale for the use of HeLa as an in vitro model has been included

Overall the manuscript is suitable for publication.

Reviewer #3 (Remarks to the Author):

The lack of enough novelty continues being an issue for this reviewer. The fact that authors cited additional references does not support an additional novelty. See previous comments: "From the viewpoint of enzyme-associated MOFs and nanometric MOFs, the novelty of this study is limited, as such composites have been subject of investigation in several works and reviews (see: T.J. Piklak et al., *Top Catal.* 2006; V. Lykourinou et al., *JACS*, 2011 ; Y. Chen et al, *Inorg Chem*, 2012; W.L. Liu et al., *Chem.*, 2014 ; D. Feng et al., *Nat. Comm.* 2015 ; P. Li et al., *ACS Nano* 2016; E. Gkaniatsou et al., *Mater Horiz.*, 2017; X. Lian et al, *Chem. Soc. Rev.* 2017, among others). Further, the association of enzymes to the PCN-333 material has already been reported by some of these authors (*Nat. Comm.*, 2015, 6, 5979). On the other site, in vitro antioxidant activity of some MOFs has been previously reported (L. Cooper et al., *Chem. Comm.* 2015; T. Hidalgo et al., *J. Mater Chem B*, 2017)."